# THE ENERGY COST OF REASONING: ANALYZING ENERGY USAGE IN LLMS WITH TEST-TIME COMPUTE

## ABSTRACT

Scaling large language models (LLMs) has driven significant advancements, yet it faces diminishing returns and escalating energy demands. This work explores how test-time compute (TTC) can serve as an energy-efficient complement to conventional scaling strategies by allocating additional computational resources at inference time rather than during training. Specifically, we investigate whether employing TTC can achieve superior accuracy-energy trade-offs compared to simply increasing model size. Our empirical analysis reveals that TTC surpasses traditional model scaling in accuracy/energy efficiency, with notable gains in tasks demanding complex reasoning rather than mere factual recall. Further, we identify a critical interaction between TTC performance and output sequence length, demonstrating that strategically adjusting compute resources at inference time according to query complexity can substantially enhance efficiency. Our findings advocate for TTC as a promising direction, enabling more sustainable, accurate, and adaptable deployment of future language models.

## 1 INTRODUCTION

Test-time compute (TTC) has emerged as a promising approach to significantly enhance the performance of large language models (LLMs) without the reliance on traditional scaling laws. Conventionally, improving neural language models involved scaling either the number of parameters or expanding the training dataset (Kaplan et al., 2020). However, this approach demands substantial computational resources, resulting in considerable energy consumption during both training and inference phases. Furthermore, such scaling is increasingly constrained by the limited availability of sufficiently large and diverse datasets(Villalobos et al., 2022), highlighting diminishing returns in terms of resource efficiency.

Recent research addresses this limitation by proposing a new strategy to enhance the performance of LLMs: allocating additional computational resources at inference time, known as TTC. TTC is inspired by human reasoning process, where humans reflect and reason logically to answer challenging questions rather than responding impulsively. Similarly, LLMs should leverage additional computational effort during inference to improve their responses. Prominent examples of TTC methodologies include prompting techniques like Chain-of-Thought (CoT) reasoning (Wang et al., 2022; Yao et al., 2023), self-revision methods (Kamoi et al., 2024; Liu et al., 2024; Madaan et al., 2023), post-training for reasoning (DeepSeek-AI et al., 2025; Zelikman et al., 2022; Singh et al., 2023), and ensemble-like approaches aggregating response from multiple parallel generations to improve the final response (Brown et al., 2024; Greenblatt, 2024; Irvine et al., 2023; Wang et al., 2022). Despite methodological differences, these strategies collectively emphasize additional computational investment during inference to enhance model accuracy.

The rising computational demands of TTC necessitate careful consideration of associated energy costs. Recent industry reports underscore this concern, with Meta attributing up to 70% of its AI power consumption to inference processes (Wu et al., 2022), Google reporting 60% of machine learning energy usage (Patterson et al., 2022), and AWS indicating inference-related demands accounting for 80-90% of computing resources (Barr, 2019) all without TTC. Moreover, broader data center energy consumption trends indicate a rapid escalation, expected to constitute up to 12% of total U.S. electricity usage by 2028 (Shehabi et al., 2024). The environmental impact and financial

implications of such intensive energy consumption underline the urgency of evaluating the sustainability and efficiency of TTC approaches.

In this paper, we examine the accuracy-energy trade-offs associated with TTC compared to traditional model scaling. Specifically, we investigate the relationship between accuracy improvements and energy consumption incurred by employing TTC methods versus scaling model sizes across multiple benchmarks. Through comprehensive experiments conducted on Qwen2.5 (Qwen et al., 2024) models (1.5B, 7B, 14B, and 32B parameters) using A100 GPUs across mathematical, coding, and common-sense reasoning benchmarks, we derive key insights into the comparative advantages and costs of TTC.

Our findings highlight several crucial observations:

- TTC demonstrates superior accuracy-energy trade-off improvements over traditional model scaling. Notably, smaller models enhanced with TTC can outperform substantially larger models relying solely on scale.
- However, indiscriminate application of TTC can drastically elevate energy consumption, with cases where TTC usage amplifies energy costs up to $113.48\times$, as observed with the 7B model on a coding benchmark.
- Output sequence length emerges as a reliable indicator of model comprehension, with extended sequences often signaling models' struggles or attempts at multiple reasoning pathways, consequently incurring higher energy usage.
- Complex questions drive models with TTC to allocate greater computational resources during inference, paralleling human cognitive processes in handling challenging tasks.

To the best of our knowledge, this study represents one of the first systematic explorations into the energy implications of TTC for LLM inference. Our analysis provides a foundational understanding of how TTC can serve as a more energy-efficient complement to traditional scaling. We explicitly focus on inference-related impacts, leaving exploration of training-phase considerations as important avenues for future research.

## 2 BACKGROUND

**LLM Inference Pipeline** Inference for transformer-based LLMs (Grattafiori et al., 2024; Qwen et al., 2024; Team et al., 2023; Achiam et al., 2023; Vaswani et al., 2017) typically involves two sequential stages: prefill and decode. Initially, when an input sequence is provided, it undergoes the prefill stage, where all tokens in the seqeunce are processed simultaneously. This stage computes the first output token and generates a Key-Value cache (KV-cache), which stores contextual information from the input sequence. Following the prefill stage, the model enters the decode stage, wherein it autoregressively generates subsequent tokens one-by-one, leveraging the previous KV-cache.

Each of these stages presents distinct computational bottlenecks. The prefill stage is predominantly compute-bound due to large matrix multiplications resulting from processing all tokens simultaneously. In contrast, the decode stage typically becomes memory bandwidth (BW)-bound because of the frequent accesses to the KV-cache.

**Test-time Compute** Recently, leveraging additional computational resources at inference time—known as TTC—has emerged as an effective strategy for enhancing the reasoning capabilities and overall accuracy of LLMs. We categorize TTC techniques into two categories based on their resource utilization patterns: 1) techniques that increase the input tokens more (Brown et al., 2024; Wang et al., 2022; Brown et al., 2020) or 2) those that increase output tokens more (DeepSeek-AI et al., 2025; Madaan et al., 2023; Zelikman et al., 2022; Singh et al., 2023). Each category uniquely affects the resource demands of the inference pipeline.

Increasing the number of input tokens primarily impacts the prefill stage as the compute requirement scales quadratically with the length of the input sequence. On the other hand, increasing the number of output tokens predominantly affects the decode stage. Specifically, extending the output length linearly increases both the size of the KV-cache and the number of autoregressive decoding steps, scaling memory BW requirements quadratically.

Table 1: List of benchmarks used in this study for Math, Code, and Common Sense tasks

| Math | Code | Common Sense |
|------|------|--------------|
| AIME24 (MAA, February 2024a) | HumanEval (Chen et al., 2021) | HellaSwag (Zellers et al., 2019) |
| AIME25 (MAA, February 2024b) | LiveCodeBench (Jain et al., 2024) | MMLU (Hendrycks et al., 2021a) |
| GPQADiamond (Rein et al., 2024) | MBPP (Austin et al., 2021) | GSM8K (Cobbe et al., 2021) |
| Math500 (Hendrycks et al., 2021b) | CodeForces (Quan et al., 2025) | CommonsenseQA (Talmor et al., 2019) |

**Energy Usage Measurement**   Early pioneering efforts, such as the study by Strubell et al. (Strubell et al., 2020), began to systematically investigate the energy cost of training transformer models, spurring subsequent studies that quantified energy consumption during model training (Patterson et al., 2022). With the increasing deployment of LLMs in practical applications, recent studies have expanded this scope to include energy measurements specific to inference workloads (Luccioni et al., 2024; Fernandez et al., 2025; Wu et al., 2025; Patel et al., 2024; Stojkovic et al., 2024).

On top of the existing works, we specifically examine the additional energy overhead incurred by TTC methods. Despite TTC becoming a widely accepted practice to improve the reasoning capabilities of LLMs, the precise energy costs associated with such approaches remain largely unexplored—highlighting the critical importance and novelty of our analysis.

## 3 METHODOLOGY

In this study, we systematically measure the additional energy consumption associated with TTC methods in the context of LLM inference. To quantify this energy overhead, we base our work on Evalchemy (Guha et al., 2024) and utilize SGLang (Zheng et al., 2024) for model execution on NVIDIA A100 GPUs (NVIDIA, 2020), each with a maximum power rating of 500W. We run each benchmark from start to end assuming that all questions are present to be processed to eliminate any scheduling overhead or queuing delay. Power consumption during model inference is monitored using the NVIDIA Management Library (NVML) (NVIDIA, 2024). We integrate power usage over the inference period to compute total energy consumption for each model evaluation.

**Categorization of TTC Methods**   We group TTC approaches into two distinct categories: (1) methods that increase the number of input tokens and (2) methods that increase the number of output tokens. For the first category, we select parallel sampling (Brown et al., 2024; Greenblatt, 2024; Irvine et al., 2023; Wang et al., 2022) with majority vote (MV) to finalize the answer. For the second category, we adopt the reasoning token (RT) approach (DeepSeek-AI et al., 2025; Zelikman et al., 2022; Singh et al., 2023). These models have demonstrated that they can reason through their answers by utilizing reasoning tokens. Unless stated otherwise, we evaluate the energy usage and accuracy of MV and RT against baseline (Base) approach which does not use any TTC methodology.

**Model Selection**   We base our study on models supported by the DeepSeek-R1 framework (DeepSeek-AI et al., 2025), selecting four representative models from the Qwen2.5 family (Qwen et al., 2024): 1.5B, 7B, 14B, and 32B variants. These models have been distilled from DeepSeek-R1 and are particularly suitable for demonstrating the effectiveness of reasoning tokens in refining model outputs. To facilitate a rigorous comparison, we also employ these same Qwen2.5 models as baseline references (without TTC) and in parallel sampling scenarios. Using identical model architectures across conditions ensures a fair and controlled evaluation. 32B models are executed across two NVIDIA A100 GPUs in tensor parallel manner as they on average use over 100GB compared to 80GB memory on our A100 GPUs.

**Evaluation Tasks**   To comprehensively capture the impact of TTC, we evaluate model performance across three representative machine learning task categories, each consisting of four distinct benchmarks. We choose mathematical reasoning and code-generation tasks to assess how effectively TTC improves a model's logical reasoning capabilities using mathematical and programming languages following existing studies (Team et al., 2023; Achiam et al., 2023; Qwen et al., 2024). Additionally, common sense tasks are selected to investigate whether extra computation at inference time can meaningfully enhance the model's retention or representation of factual information. Detailed descriptions of tasks and corresponding datasets are summarized in Table 1.

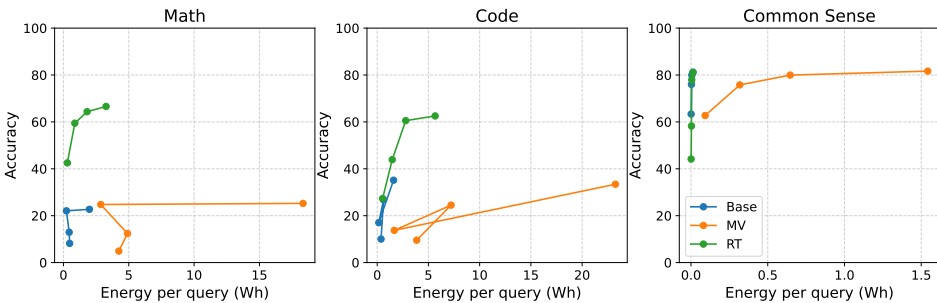

Figure 1: Accuracy versus energy per query averaged across four benchmarks in each task. Each dot in the line represent Qwen2.5 1.5B, 7B, 14B, and 32B from left to right, respectively.

## 4 RESULTS

We present our results on Base, MV, and RT. Evaluations are run 10 times with batch size 16, maximum sequence length 32768, and data type Bfloat16 unless stated otherwise. We generate 5 samples when running MV with parallel sampling. Common-sense benchmarks take roughly 10 minutes per evaluation while math and code benchmarks take between 1 to 4 hours.

### 4.1 OVERVIEW

**Accuracy vs Energy per Query**   The primary objective of this analysis is to explore the trade-off between accuracy improvements and associated energy costs when employing TTC. Figure 1 presents this trade-off across mathematical, coding, and common-sense benchmarks.

Firstly, the slope of the accuracy-energy curve is steeper for math and common-sense tasks, with a clearly defined knee point situated higher and further from the origin for RT. This indicates a superior accuracy-to-energy ratio when using RT. For example, in mathematical benchmarks, increasing the model size from 1.5B to 7B parameters in the Base configuration yields only a 4.8% accuracy improvement with negligible energy difference per query. In contrast, using RT on the 1.5B model significantly boosts accuracy by 34.3%, representing a 7.5-fold increase in accuracy improvement over traditional scaling, at similar energy consumption levels.

In coding benchmarks, scaling model size offers better accuracy/energy gains at lower model scales than RT. Specifically, transitioning from a Base 1.5B to Base 7B model increases accuracy by 16.8% while reducing energy consumption per query by 40.1%. Conversely, applying RT to the 1.5B model achieves a similar accuracy improvement (17.3%) but increases energy per query by 57.4%. However, when targeting higher accuracy regimes (e.g., Base 32B model versus RT 7B model), RT demonstrates a superior slope of accuracy improvement per unit energy consumed (28.1%/Wh for RT versus 20.7%/Wh for Base).

Common-sense benchmarks show limited or even adverse performance when employing RT. Specifically, the Base 1.5B model outperforms both the 1.5B and 7B RT configurations. This is expected because common-sense tasks predominantly evaluate factual knowledge retrieval rather than complex reasoning capabilities, limiting the effectiveness of RT.

Secondly, the use of MV combined with parallel sampling substantially increases energy consumption without proportionate accuracy improvements. This inefficiency occurs because MV relies on simplistic aggregation, where the probability of correctness, $p$, remains unchanged. Particularly, if $p < 0.5$ and we sample 5 generations, using MV with multiple candidates can potentially reduce accuracy due to the binomial distribution of generations. Consequently, a more efficient aggregation method is necessary to better utilize candidate outputs and justify increased energy consumption.

Finally, an intriguing anomaly emerges within the coding benchmarks: Base and MV configurations for the 14B models exhibit lower accuracy compared to their 7B counterparts. This finding reinforces the notion that merely increasing model size does not guarantee enhanced accuracy if underlying reasoning capabilities remain insufficient.

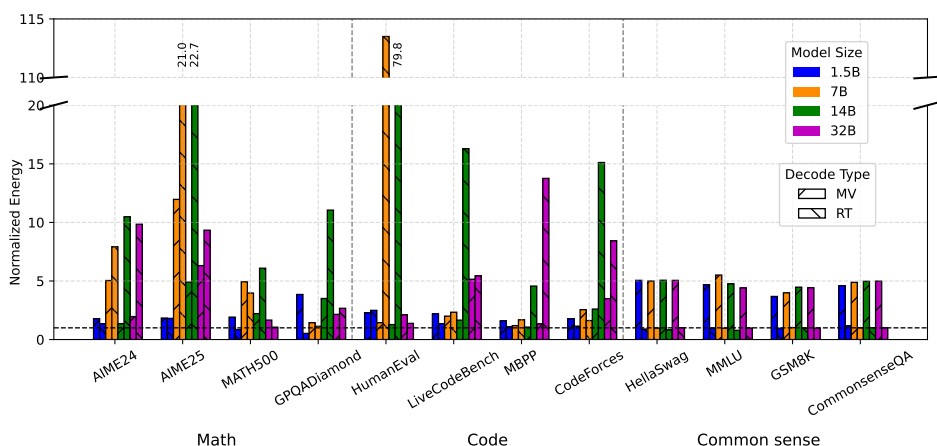

Figure 2: Energy consumption of MV and RT normalized to Base which does not use TTC. Left bars in each color represent MV and right bars represent RT. The first, second, and third sets of four benchmarks represent math, code, and common sense, respectively. The dotted horizontal line represents Base. Note that the y-axis is cut off from 20 to 110.

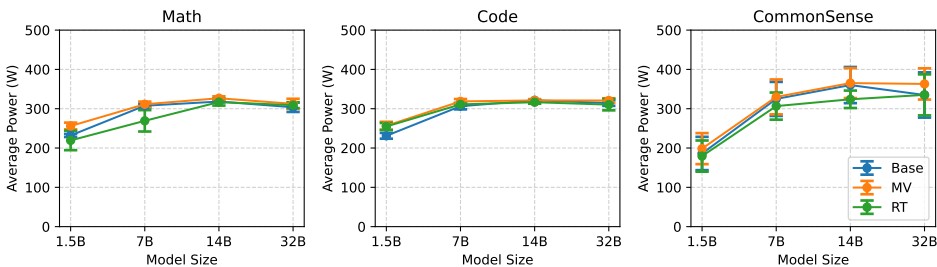

Figure 3: Power readings during runtime averaged across four benchmarks in each task.

**Normalized Energy** We analyze the energy consumption implications of employing TTC, specifically quantifying the energy increase introduced by MV and RT relative to Base. Figure 2 illustrates these normalized energy metrics.

On average, MV consumes $2.63\times$, $2.01\times$, and $4.69\times$ more energy than the Base configuration for mathematical, coding, and common-sense benchmarks, respectively. RT demonstrates a more substantial average energy increase, using $3.66\times$, $10.4\times$, and $1.08\times$ more energy than Base for math, code, and common-sense tasks, respectively.

The energy cost of RT, in particular, can escalate dramatically, with a peak increase of $113.48\times$ for the 7B model on the HumanEval coding benchmark. This extreme rise is largely driven by a substantial growth in the number of generated tokens; RT models produce, on average, $4.4\times$ more tokens than Base, and in extreme cases up to $46.26\times$ more tokens.

In contrast, although MV multiplies both input and output tokens by a factor of five, the short output sequences do not fully utilize GPU compute and memory BW during decoding. Consequently, this allows MV to maintain minimal latency overhead despite an increased batch size, resulting in less than a proportional $5\times$ increase in energy consumption. However, common-sense benchmarks, which generate only a single output token correlating the answer to multiple-choice options, consistently exhibit approximately $5\times$ energy usage relative to Base.

**Power Measurements** We explore power consumption implications of employing TTC. Data centers typically have strict power provisioning limits, requiring accurate assessments of power usage to prevent exceeding provisioned capacity. Figure 3 provides average GPU power measurements and associated standard deviations across each benchmark.

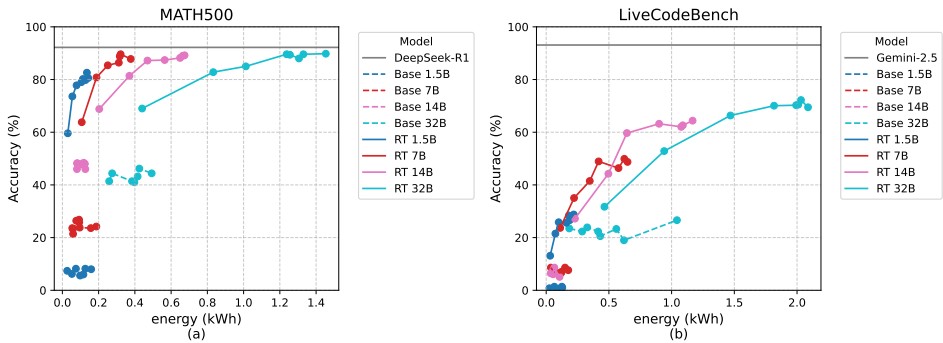

Figure 4: Energy vs Accuracy per length. Dotted and solid lines represent Base and RT, respectively and each color represent different model size. Each of the ten dots on a line represent output sequence length limit starting from one-tenth of the maximum sequence length of a model to the maximum sequence length. Grey lines on the top represent the best models at the time of writing.

Our analysis reveals that TTC only marginally affects power usage. Specifically, MV introduces an average increase of approximately 5% in power consumption due to increased batch sizes from parallel sampling. Conversely, RT tends to use slightly less power than Base. This unexpected reduction occurs because RT inference is predominantly decode- and especially attention layer-bound, characterized by extremely long output sequences. The resulting high memory requirements for the KV-cache create a bottleneck that saturates GPU memory BW but significantly under-utilizes the GPU compute units.

For instance, RT benchmarks produce an average of 7845 tokens per query, translating to an average KV-cache size of about 258MB, even for relatively small (7B) models. Given the memory BW saturation threshold (approximately 100MB data array for A100 GPUs (Anzt et al., 2020)), the inference process becomes memory-bound, limiting the effective utilization of GPU computational resources. Consequently, the power usage stabilizes at a level consistent with these memory constraints.

## 4.2 IN-DEPTH ANALYSIS

We conclude that parallel sampling with MV shows lower accuracy/energy than RT and that TTC has limited benefit on common-sense benchmarks. To this end, we provide a thorough analysis on RT using one reasoning benchmark each from math (MATH500) and coding (LiveCodeBench) task.

**Length Sweep** We investigate how the maximum output sequence length affects RT performance and associated computational costs. We incrementally limit the output sequence length by steps of 3277 tokens which is one-tenth of the original maximum sequence length.

Figure 4 demonstrates the relationship between accuracy and energy consumption for two representative benchmarks. Notably, RT consistently forms a distinct and superior Pareto frontier compared to Base models. This distinction arises because Base models reach an accuracy ceiling even when allowed extended sequence lengths, with improvements requiring substantially larger model sizes. Base accuracy saturates at 48.3% for MATH500 and 23.5% for LiveCodeBench, whereas RT achieves significantly higher accuracy ceilings of 89.8% and 72.1%, respectively. Moreover, the RT Pareto frontier highlights that slight accuracy trade-offs can yield considerable energy savings, valuable in resource-constrained environments. For example, transitioning from RT 32B to RT 14B to solve LiveCodeBench reduces energy consumption by 1.74× while sacrificing only 7.8% accuracy.

Additionally, RT models often terminate computations before reaching the maximum sequence length, which can be seen from the clustered points at the end of all RT lines. This indicates their ability to avoid excessive processing when uncertain, thus efficiently managing resources.

Figure 3(a) reveals the competitive potential of smaller RT models. For instance, the 1.5B RT model achieves 83.2% accuracy with just 135.22Wh, outperforming the Base 32B model at only 45.7% accuracy and consuming 424.54Wh. Remarkably, the RT 7B model reaches an accuracy of 89.6%, approaching the performance of much larger models such as DeepSeek-R1 with 617B parameters.

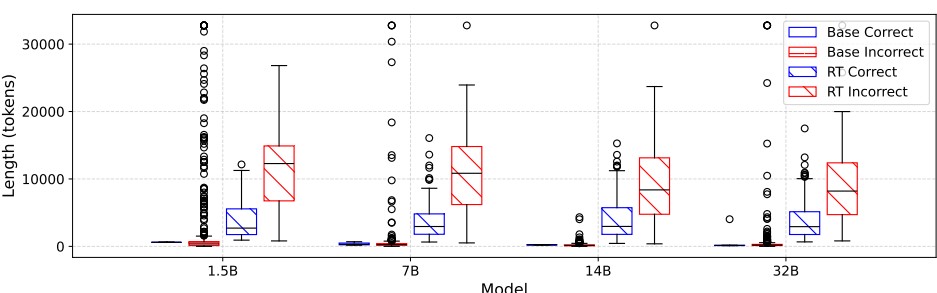

Figure 5: Output token count distribution of correct and incorrect queries.

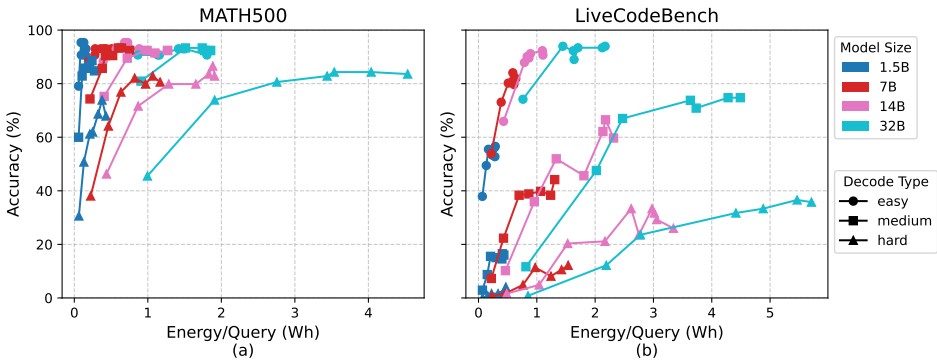

Figure 6: Accuracy vs energy per difficulty level. Each dot on a line represents different output sequence length limit identical to the previous Length Sweep analysis.

**Tokens per Correct and Incorrect Queries**   Figure 5 illustrates the number of output tokens generated for correct and incorrect answers in both Base and RT models highlighting how sequence length relates to model reasoning.

Base models exhibit clustered sequence lengths for both correct and incorrect answers, with numerous outliers among incorrect responses. These models' limited reasoning capabilities result in limited thinking process and outlier sequences typically represent unnecessary token generation rather than deeper reasoning. Increasing the model size reduces the number of outliers but maintains clustering near short sequence lengths, underscoring their restricted reasoning abilities.

Conversely, RT models utilize fewer tokens to reach correct answers compared to incorrect responses. For correct queries, token length distributions are consistent across model sizes, suggesting that RT effectively guides reasoning up to a certain point. However, when uncertain, RT models produce longer incorrect responses due to loops or divergent reasoning. Larger models significantly reduce the token lengths of incorrect answers, indicating a stronger capability to recognize ineffective reasoning paths and terminate more efficiently.

**Query Difficulty**   We analyze the impact of task difficulty on energy consumption and accuracy using RT. Figure 6 depicts the accuracy versus energy-per-query tradeoff across easy, medium, and hard difficulty levels for selected benchmarks.

Firstly, task difficulty directly correlates with increased token generation and higher energy consumption, with each difficulty level forming distinct Pareto frontiers. This indicates alignment between model-perceived and human-perceived task difficulty, where additional computational effort reflects increased complexity. This observation expands on previous findings that models' and humans' perceptions of difficulty align, especially for reasoning-based tasks.

Secondly, smaller models with RT can match the performance of larger models for easier tasks, resulting in significant energy savings. Specifically, using 14B and 1.5B models instead of 32B models saves $11.55\times$ and $1.48\times$ energy for easy tasks in MATH500 and LiveCodeBench, respectively.

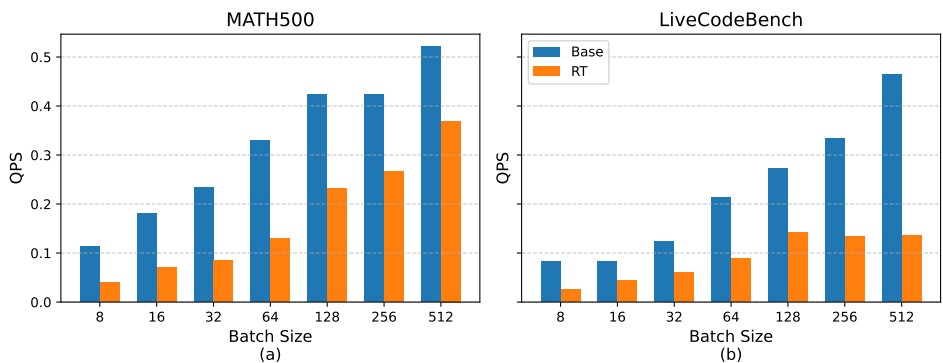

Figure 7: Throughput vs Batch size.

Further analysis reveals that using RT provides better accuracy-per-energy improvements than simply increasing model size, even when all models employ RT. For example, on the hard queries of MATH500, the 14B model increases accuracy from 46.2% to 71.6% at an additional 0.4Wh, whereas scaling to a 32B model has marginal impact on accuracy while using 0.6Wh more energy. Similarly, the 14B model for medium-difficulty tasks in LiveCodeBench improves accuracy from 10.0% to 36.2% with an additional 0.96Wh of energy, demonstrating an accuracy/energy benefit of 39.5%/Wh compared to only 3.33%/Wh when scaling up to the 32B model. Thus, RT consistently offers superior or equivalent accuracy-per-energy tradeoffs compared to increasing model sizes.

**Impact of Batch Size on Throughput**   We examines the influence of batch size on throughput (queries-per-second, QPS) of RT compared to Base. Increasing batch size is a common strategy to enhance throughput, particularly important given the increased latency and reduced throughput associated with the longer output sequences generated by RT.

Figure 7 depicts the QPS across various batch sizes for the two benchmarks using the highest-performing 32B models. Results indicate that larger batch sizes consistently improve throughput for both configurations. For Base, output sequences are relatively short, resulting in smaller KV-cache footprints that under-utilize GPU memory BW. Although RT applied to MATH500 generates more tokens than Base, it still generates short enough sequences to under-utilize GPU memory BW. As a result, even at high batch sizes, the KV-cache remains manageable, and throughput scales proportionally with batch size increases.

However, RT applied to the LiveCodeBench exhibit a notable saturation in throughput at a batch size of approximately 128. On average, RT generates around 6588 output tokens for LiveCodeBench, shifting inference into a predominantly decode-bound stage. During the decode stage, loading the large KV-cache that scales linearly with batch size from GPU memory becomes the critical performance bottleneck, eventually saturating available GPU memory BW.

**Impact of LLM Serving System Optimizations**   The most relevant optimization in our study is prefix caching, which stores the KV-cache of a query and reuses it whenever the same prefix appears in later queries. As shown in Table 2, prefix caching offers little energy savings for RT, since RT does not use additional tokens during the prefill stage. MV shows a more nuanced picture as prefix caching does reduce energy consumption, but less than expected. In theory, the normalized energy consumption for MV should approach 1 if queries are dominated by the prefill stage. However, we find that MV generates at least one sample with an output sequence an order of magnitude longer than the other samples. This costly event shifts the bottleneck from prefill to decode, making the decode stage the primary driver of energy usage in the benchmarks we evaluate.

System optimizations targeting the decode stage such as speculative decoding (Leviathan et al., 2023; Li et al., 2024; Chen et al., 2023; Kim et al., 2023) benefits RT and MV with prefix caching more than Base. Speculative decoding reduces the cost of the decode stage for TTC and Base by the same factor (Leviathan et al., 2023) by using a smaller model to generate multiple tokens and calling an original model only to verify the tokens. The overall benefit for MV and RT is larger than that for

Table 2: Energy consumption of MV and RT with and without the use of prefix caching normalized to Base. Normalized energy consumption for RT stays similar as no tokens were added during the prefill stage. Prefix caching saves energy consumption for MV but smaller than anticipated.

| | MV | | RT | |
| Tasks | Prefix Cache | No Prefix Cache | Prefix Cache | No Prefix Cache |
|---|---|---|---|---|
| Math | 1.91 | 2.28 | 2.11 | 2.37 |
| Code | 2.45 | 3.52 | 4.85 | 4.94 |
| Common Sense | 4.96 | 4.75 | 1.01 | 1.01 |

Base as the decode stage is more dominant than the prefill stage for the two TTC approaches. Another major proposal disaggregated serving (Zhong et al., 2024), separates prefill and decode across different clusters. This approach allows systems to reclaim idle resources, but incurs overhead from transferring the KV-cache between clusters. Importantly, just like speculative decoding, disaggregated serving applies its costs and savings uniformly to TTC and Base, maintaining the trend that we report in this study.

## 5 CONCLUSION AND DISCUSSION

In this work, we systematically evaluate the energy consumption of large language model (LLM) inference specifically through Test-Time Computation (TTC). We investigated two prevalent TTC strategies: (1) generating multiple candidate answers, and (2) self-refinement through iterative reasoning. Our analysis reveals that employing TTC strategies generally achieves better accuracy-energy trade-offs than solely increasing model size. Additionally, output sequence length emerges as a reliable indicator of model comprehension and complex questions require greater computational resources during inference. Combining larger models with TTC strategies yields the highest accuracy but comes with substantial energy costs. Thus, a prudent approach is essential, balancing accuracy gains against energy expenditures. Our findings suggest promising directions for leveraging TTC in a more energy-efficient manner.

**Difficulty-aware Model Selection** Our results (Figure 4) demonstrate that smaller models with TTC can compete effectively against larger models, with distinct Pareto frontiers emerging for each task difficulty level. This insight supports the implementation of difficulty-aware evaluators (OpenAI, 2025; Säuberli & Clematide, 2024; Park et al., 2024; Dutulescu et al., 2024; Xu et al., 2024) that dynamically route queries to the most energy-efficient model achieving optimal accuracy. Optimal energy consumption $E_o$ of an oracle evaluator and realistic energy consumption $E_r$ of an imperfect evaluator in practical settings employing a high-performing model as a fallback can be calculated as

$$E_o = \sum_{i=0}^{D} n_i e_{m_i} \qquad (1a) \qquad\qquad E_r = \sum_{i=0}^{D} p_i N e_{m_i} + q N e_{m_D} \qquad (1b)$$

where $p_i$ denotes the probability that the evaluator correctly predicts difficulty level $i$, $q$ represents the probability of incorrect predictions, and $e_{m_D}$ is the energy consumed by the largest, most accurate model. Compared against the baseline approach ($E_{max} = N e_{m_D}$), our analysis shows significant potential energy savings. Systems with an oracle evaluator consume 1.75 kWh compared to baseline consumptions of 2.03 kWh and 2.20 kWh, for LiveCodeBench and MATH500, respectively.

**Length-wise Early Exit** Figure 5 indicates that correct responses typically require fewer output tokens than incorrect responses, suggesting another promising avenue for efficiency improvements. Existing early-exit methods focus on exiting at certain model layers (Del Corro et al., 2023; Chen et al., 2023; Teerapittayanon et al., 2016; Zhou et al., 2020). Extending these techniques to include token-length threshold will enable models to cease generation when answers become evidently flawed or excessively verbose. Such a mechanism effectively reduces unnecessary energy consumption. Specifically, Figure 5 illustrates that the correct answers use less than 12000 tokens excluding outliers. We apply this observation on Figure 4, cutting the generation at the fourth point (13108 tokens) on the RT lines. This gives us 14% accuracy saving without loss of accuracy.

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
