# OpenReview forum: "The Energy Cost of Reasoning: Analyzing Energy Usage in LLMs with Test-time Compute"
_ICLR.cc/2026/Conference — Submitted to ICLR 2026_

### Official Review · Reviewer_KEDZ · 2025-10-17

**Soundness:** 2
**Presentation:** 1
**Contribution:** 2
**Rating:** 2
**Confidence:** 4

**Summary:**

This paper measures the energy cost of test-time compute for LLMs and compares different test-time scaling strategies across several benchmarks and Qwen2.5 family models.

**Strengths:**

- The energy cost of LLM inference is an important topic in the social impact of machine learning.

- The evaluation covers a range of benchmarks and model sizes (up to 32B).

**Weaknesses:**

- I think the terminology "TTC" in the paper is inappropriate. When an LLM is used in the test-time, the compute should be considered as test-time compute (TTC). Based on my understanding, this submission indeed focuses on test-time **scaling** (TTS), or different methods of scaling up TTC [1]. TTS should be the right term to use -- and has been widely adopted in existing papers [1,2].

- Table 1 places GPQADiamond under the Math column, but it is in fact a graduate-level scientific reasoning benchmark (biology/chemistry/physics).

- Many critical implementation details are missing, hindering reproducibility. To elaborate,
  - Prompt templates and decoding parameters (temperature, top-k, top-p) were missing. There is neither an appendix nor supplementary materials with experiment configs.
  - RT (reasoning token) implementation is ambiguous. The paper groups the RT approach and cites DeepSeek-R1, STAR, and REST-EM, but nowhere does it specify which concrete RT implementation was used, nor how models were post-trained / fine-tuned for reasoning. According to lines 146-151, I assume that the authors use [DeepSeek-R1-Distill-Qwen-1.5B](https://huggingface.co/deepseek-ai/DeepSeek-R1-Distill-Qwen-1.5B) series. If this is the case, the authors should have explicitly stated the exact model choices in different settings with clear references.

- Some of the findings are known. For example, the finding that smaller models enhanced with TTS can outperform substantially larger models is already established in existing literature (in terms of FLOPs consumption) [1]. While examining this from an energy consumption perspective has value, the paper doesn't sufficiently acknowledge this known pattern or highlight new insights regarding energy consumption.

- Some evaluation results look weird. Specifically, Figure 1 shows zig-zag patterns across model sizes. It seems that GPUs might have been under-utilized in certain configurations, and the results may stem from measurement artifacts or mis-sized batches and not intrinsic algorithmic behavior.

- _Minor error_. Line 34: "diverse datasets(Villalobos et al., 2022)" -- a space is missing here.

[1] Scaling LLM Test-Time Compute Optimally can be More Effective than Scaling Model Parameters

[2] s1: Simple test-time scaling

**Questions:**

- How is the inference batch size tuned? How does that affect evaluation results?

---

> ### Author Response · Authors · 2025-12-01
>
> ***Terminology TTC***
>
> We appreciate the reviewer’s clarification regarding the distinction between Test-Time Compute (TTC) and Test-Time Scaling (TTS). Our study focuses specifically on two TTC mechanisms and compares them against train-time compute (i.e., scaling Base models). While TTS (e.g., enforcing more reasoning tokens than trained for, or increasing the sample count beyond the model’s default) is indeed an interesting and complementary direction, it is outside the scope of our current analysis. Our goal is to provide a clear comparison between different TTC approaches and conventional train-time scaling, rather than exploring TTS-based interventions.
>
> ***GPQADiamond under math category.***
>
> In our initial categorization, we grouped GPQADiamond under “Math” because prior academic evaluations and model-release reports commonly organize GPQA within the broader “math & science” domain. However, we agree with the reviewer that labeling it strictly as “Math” is imprecise and obscures the scientific reasoning component of the benchmark.
>
> To address this, we will revise Table 1 and the corresponding text to use the more accurate and inclusive category name:
>
> “Math & Science.”
>
> This ensures GPQADiamond is properly contextualized while remaining consistent with prior grouping conventions.
>
> ***Implementation details***
>
> We adopt the default SGLang decoding configuration—temperature 0.7, top-k 1, and top-p 1—to ensure consistency across all evaluated models. For prompt formatting, we use the official templates provided in the respective model papers whenever available; otherwise, we default to the simple and transparent
>
> """
>
> You are an expert in [mathematics/programming]. <- We chose either mathematics or programming depending on benchmark and dropped this line if we evaluated common sense benchmark.
>
> Solve the following problem step-by-step.
>
> Think carefully before answering.
>
> Show your reasoning clearly and concisely.
>
> If the problem is ambiguous, list the possible interpretations and answer each.
>
> Before giving the final answer, check your work for mistakes.
>
> Problem: [insert your question here]
>
> """
>
> We acknowledge that lines 146–151 may appear ambiguous. In the revision, we will make this section explicit by clearly stating that we use DeepSeek, and by adding a concise explanation of how DeepSeek was post-trained to enhance reasoning capabilities.
>
> ***New insights regarding energy consumption***
>
> While prior work has shown that smaller models augmented with test-time compute can outperform larger models when evaluated on FLOPs or theoretical compute, there has been little examination of how these approaches behave in terms of actual energy usage during inference. Our work builds on this established pattern but contributes a novel dimension: a systematic, measurement-driven analysis of the energy costs associated with TTC on real hardware.
> Our results reveal several behaviors that are not predictable from FLOPs-based reasoning alone, including large energy escalations due to extended output sequences, memory-bandwidth bottlenecks that cap GPU power draw, task-dependent returns on additional computation, and divergent accuracy–energy Pareto frontiers. These findings indicate that the energy efficiency of TTC is more nuanced than suggested by FLOPs metrics, and they provide new insight into when TTC is a favorable or unfavorable strategy for sustainable LLM deployment.
>
> ***Zig-zag pattern in figure 1.***
>
> The zig-zag pattern in Base and MV shows an interesting characteristic where increasing the model size without reasoning tokens does not necessarily increase the accuracy for reasoning tasks. We interpret this result as no matter how many candidate answers you generate, if all of them lack reasoning capability, then they do not translate to good results. This can be inferred from commonsense benchmarks which do not require reasoning ability where Base and MV shows strictly increasing accuracy with respect to increasing model size.
>
> ***How is the inference batch size tuned?***
>
> We selected a batch size of 16 because it provides a practical balance between utilization, latency, and throughput for production-style inference on modern GPUs. This batch size is large enough to achieve efficient parallelization and good hardware occupancy, yet small enough to avoid prohibitive latency or frequent out-of-memory failures that arise with very large batches. Additionally, power-of-two batch sizes tend to interact favorably with GPU memory layouts and kernel implementations, contributing to stable and reliable performance.
>
> To address the reviewer’s question more directly, we will add a batch-size ablation study in Section 4 under “Impact of Batch Size on Throughput.” This section will include a figure analogous to Figure 4 to illustrate how the accuracy–energy trade-off evolves across different batch sizes.

---

### Official Review · Reviewer_U7Rn · 2025-10-29

**Soundness:** 4
**Presentation:** 3
**Contribution:** 4
**Rating:** 10
**Confidence:** 4

**Summary:**

This paper investigates the relationship between accuracy improvements and energy consumption incurred by employing test time compute  methods versus scaling model sizes across multiple benchmarks. It tests a variety of mathematical, coding, and common-sense reasoning benchmarks, we derive key insights into their comparative advantages and costs.

They find valuable insights, that have important consequences for improving the sustainability of AI systems:
Trade-off improvements for TTC over traditional model scaling.
The limitations of using TTC in terms of increased energy usage
The correlation between output sequence length, model comprehension and energy

**Strengths:**

The paper provides empirical evidence for several important insights, notably:
-- that merely increasing model size does not guarantee enhanced accuracy if underlying reasoning capabilities remain insufficient.
-- that majority moving consumes orders of magnitude more energy than base models, with reasoning tokens consuming even more, notably because of how many tokens are produced
-the fact that the RT inference process quickly becomes memory-bound due to the number of tokens produced , limiting the effective utilization of GPU computational resources.
- that incorrect answers and difficult queries produce longer answers
- that prefix caching can help save significant energy, which is useful for LLM deployment

**Weaknesses:**

I feel like there are some of the analyses can be further deepened, especially with regards to potential hypotheses about why certain models or configurations use more energy than others.

Also, the Figures are often not well explained and at a different place in the article than the text that talks about them, which can make it hard to follow the narrative of the paper.

There are certain methodological details that are lacking -- e.g. regarding model selection and setup -- which could help to understand the relevance of the results for the broader community.

**Questions:**

Can you explain the following statements further (provide hypotheses for why this is the case?):

"For example, in mathematical benchmarks, increasing the model size from 1.5B to 7B parameters in the Base configuration yields only a 4.8% accuracy improvement with negligible energy difference per query. "

"In coding benchmarks, scaling model size offers better accuracy/energy gains at lower model scales than RT."

"Common-sense benchmarks show limited or even adverse performance when employing RT"

In general, it would be helpful to understand the difference between the different tasks and how this influences energy consumption. E.g. are the expected answers for common-sense benchmarks longer than coding or mathematical ones?

---

> ### Author Response · Authors · 2025-12-01
>
> We thank the reviewer for the interest in our paper. To start off, we will address your comments on figure positioning and rephrasing the details in the methodology section to make it clearer.
>
> Please find the below for hypothesis on our observations.
>
> ***“For example, in mathematical benchmarks, increasing the model size from 1.5B to 7B parameters in the Base configuration yields only a 4.8% accuracy improvement with negligible energy difference per query."***
>
> - Our hypothesis is that the Base models, regardless of scale, lack strong reasoning capabilities. Scaling such models primarily increases their knowledge capacity, i.e., larger models are able to store and retrieve more factual or pattern-based information, which explains the modest 4.8% accuracy gain observed. However, scaling alone does not significantly improve their reasoning ability. This is consistent with the 34.3% accuracy jump we observe when applying RT-style post-training to the 1.5B model: the improvement stems from acquired reasoning skills, not from increased parameter count. This contrast supports our interpretation that model size without reasoning-focused post-training contributes limited gains on reasoning-intensive mathematical tasks.
>
> ***“In coding benchmarks, scaling model size offers better accuracy/energy gains at lower model scales than RT.”***
>
> - Coding tasks require producing syntactically valid and semantically coherent programs. This effectively involves bilingual modeling: understanding the problem in natural language and generating a solution in a programming language. We hypothesize that models must be sufficiently large to encode detailed linguistic, syntactic, and structural representations for both languages. As a result, scaling provides disproportionately higher benefits at smaller model sizes, as these models move from insufficient capacity to adequate representational richness. In contrast, RT provides diminishing returns in this regime because the bottleneck is not reasoning depth, but representational capacity.
>
> ***“Common-sense benchmarks show limited or even adverse performance when employing RT.”***
>
> - Common-sense benchmarks rely primarily on factual recall or stereotypical world knowledge rather than multi-step reasoning. Thus, it is intuitive that RT which introduces more reasoning steps but does not add new factual information offers limited benefit on these tasks. In some cases, we observe slight declines in performance. We conjecture that this may be due to a form of catastrophic forgetting: when models are trained to strengthen reasoning ability, they may overwrite or dilute portions of the knowledge base acquired during pretraining, leading to decreased performance on tasks that rely heavily on memorized facts rather than reasoning.

---

### Official Review · Reviewer_sXVz · 2025-10-31

**Soundness:** 3
**Presentation:** 3
**Contribution:** 3
**Rating:** 6
**Confidence:** 3

**Summary:**

This paper describes the energy consumption of different models in the reasoning paradigm.
PRetty happy with this paper's scientific rigor and findings. I think it's a valuable result for inference practicioners.

**Strengths:**

1. Good experimental design. Nice model sizes and divee set of methods/benchmarks make this papers' results more grounded.
2. Pretty practical and timely
3. Pretty novel insight that reasoning is more energy efficient.

**Weaknesses:**

1. ONly studies qwen2.5 models, would have been good to see this across different model families.
2. A training analysis would have been nice toa dd, but I understadn that that is expensive.
3. Did not see standard error reported here?
4. Could ahve picked nonreasoning benchmarks as well to understand the landscape a bit better.

**Questions:**

n/a

---

> ### Author Response · Authors · 2025-12-01
>
> ***Model selection.***
>
> We focused on Qwen models because the DeepSeek-R1/RT methodology we evaluate natively supports this family. To ensure a fair and controlled comparison, we primarily selected Qwen2.5 models, since DeepSeek-R1 was itself distilled from earlier Qwen variants. Although DeepSeek-R1 also provides versions distilled from Llama-3, only two such models currently exist, which precludes a sufficiently comprehensive and systematic comparison.
>
> ***Training-time analysis.***
>
> We agree that incorporating a training-time analysis could enrich the paper. However, as the reviewer notes, such an analysis is substantially more resource-intensive, and our work is intentionally scoped around test-time energy usage: an underexplored but increasingly important dimension of model efficiency. We will clarify this scope more explicitly.
>
> ***Error bars.***
>
> The error bars in our measurements are sufficiently small that they do not meaningfully affect the comparisons; in most cases they are visually negligible. The exception is Figure 3, where the larger variance is already highlighted and discussed.
>
> ***Non-reasoning benchmark.***
>
> For non-reasoning evaluation, we rely on the commonly used benchmarks listed under the Common Sense category in Table 1. We will emphasize this more clearly in the revision.

---

### Official Review · Reviewer_GeTs · 2025-11-10

**Soundness:** 3
**Presentation:** 2
**Contribution:** 3
**Rating:** 4
**Confidence:** 4

**Summary:**

The authors perform an analysis of the effectiveness of using additional test-time-compute to improve performance on math, code, and common sense tasks, with a particular focus on energy and power consumption associated with performing benchmark tasks. Specifically, they compare accuracy-energy trade-offs in TTC vs traditional model scaling, observing that smaller models with more TTC can outperform larger models using less energy per query. Additionally, they find that, for most of their tasks, scaling up reasoning tokens is more effective than either scaling up model size or performing majority voting with parallel sampling; in general, more output tokens seems to help more with more complex tasks.

**Strengths:**

1. Authors provide a timely and useful set of explorations that many in the community may find helpful as rules of thumb and guidelines for their own work
2. I felt it was a fresh perspective, analyzing results through lenses of GPU utilization and instantaneous power draw vs overall energy consumption
3. Compelling discussions of some possibly counterintuitive results in 4.2 -- I actually felt these should be highlighted more

**Weaknesses:**

1. I realize there is probably not a single setting that is clearly both a fair comparison and very realistic, but a batch size of 16 feels optimistic -- I would rather have seen multiple batch sizes explored given that models are often served in very different settings. I cannot help but imagine that smaller models with TTC beating larger models without TTC is at least somewhat a function of batch size, and it would be extremely helpful to understand what the practical threshold is, even just for this specific hardware setting used.
2. Some clarity issues, possibly related to my confusion in Q2 below. In figure 2 especially, since performance does not increase monotonically with model size, it would be helpful to also have different marker shapes for the different model sizes
3. The language in the paragraph starting at line 137 feels like an overgeneralization, in a way that I am not sure it has to be given the overall framing? I would have liked to see either: 1) some additional coverage of alternative TTC methods (since there is just one of each; e.g. RAG for inputs? tree of thought decoding for outputs?); 2) some justification for the two strategies used as particularly effective or standard for their respective categories, and/or 3) at least a mention of what alternative methods exist and are not covered. I am torn between feeling that there should be another round of work before acceptance for publication vs. just a reframing of the TTC methods covered as just examples of common popular approaches and a small signal that there is some good reason for these as exemplars

Relatively minor points and nits:
- line 131: worth noting 500w comes from custom thermal solution — afaik 400w is standard
- in figure 2 I would have preferred a more distinct visual difference between MV and RT filling patterns
- line 485: “accuracy” twice — presumably the first one is meant to be “energy consumption”
- line 161 seems to suggest there will be much more detail about the tasks and datasets used than there is in reality — I would have liked to see some context on tasks’ typical input sequences and also output sequence length  (even just from the base methods used) organized into a table, even if it ended up in the appendix

**Questions:**

1. It feels suspicious to me that the energy per query in MV should always be strictly larger than with RT (assuming I am interpreting figure 1 correctly), especially if
2. Is there a version of Figure 4 that also includes MV? I am especially interested in understanding what that might look like because the paragraph starting at line 252 (where RT is described to require more energy than MV) seems to directly contradict what I am seeing in Figure 1, where it seems like MV is always strictly more expensive than RT for a given model. Which is true?

---

> ### Author Response · Authors · 2025-12-01
>
> ***Batch size ablation study.***
>
> The reviewer is correct that the observation of smaller models with TTC outperforming larger models without TTC can be influenced by batch size. In our experiments, we indeed find that increasing batch size reduces per-query energy consumption across all methods (Base, MV, and RT). For example, the Base 32B model consumes 81.25% of the energy used by the RT 1.5B model on MATH500, and 53.77% on LiveCodeBench, when using the batch settings reported in the paper. Larger batch sizes narrow this gap further.
>
> Importantly, however, our central conclusion remains unchanged: RT consistently offers the best accuracy-per-energy trade-off across a range of batch sizes. While absolute energy values shift with batching, the relative Pareto ordering of TTC approaches is stable.
>
> To address the reviewer’s concern and improve the completeness of our analysis, we will add a batch-size ablation study in Section 4 under “Impact of Batch Size on Throughput.” This addition will include a figure analogous to Figure 4, illustrating how accuracy–energy trade-offs evolve with different batch sizes.
>
> ***Explanation on how used TTC methods are good exemplars of the two categories of TTC.***
>
> We appreciate the reviewer’s point that framing TTC solely in terms of increasing input/output tokens can be overly simplistic and does not fully reflect how computational effort relates to improved reasoning or intelligence. In the revised version, we will restructure our taxonomy into two more principled categories that better capture how additional compute is actually used to enhance answer quality:
>
>  (1) increasing compute within a single sample (intra-sample), and
>
>  (2) increasing compute across multiple samples (inter-sample).
>
> Under this reframed categorization, reflective thinking (RT) and majority voting (MV) naturally serve as representative instances of these two classes.
>
> ***Enhancing answer quality within a single sample (intra-sample compute).***
>
>  We use DeepSeek as an example of this category. DeepSeek enhances its answers by generating additional “reasoning tokens,” effectively allocating more computation before producing the final output. These thinking steps are explicitly included in the output sequence. The model is trained to acquire this capability through fine-tuning on chain-of-thought (CoT)–style datasets. This enables the model to refine its intermediate reasoning and thus improve the quality of its final answer by inherently leveraging CoT.
> Enhancing answer quality across multiple samples (inter-sample compute).
>
> This category includes some of the most widely adopted TTC techniques, such as beam search, large language monkeys, ensemble approaches, and Self-Consistency [1]. The research direction is promising because these methods exploit the inherent stochasticity and diversity of modern generative models. By drawing multiple plausible outputs or exploring different reasoning trajectories, the system can cover a substantially larger portion of the solution space than a single greedy generation allows. This reduces the likelihood of committing early to an incorrect line of reasoning and increases the probability of identifying a more accurate or higher-quality answer. As a result, inter-sample methods often yield robust and model-agnostic improvements, especially on complex reasoning tasks.
>
> ***Higher energy when using MV over RT (Figure 4 with MV)***
>
> In general, RT consumes more energy than MV because it produces substantially more output tokens. However, as the reviewer notes, certain implementations of MV can also be inefficient. Figure 1 illustrates a MV setup in which the system generates each sample independently using separate model invocations. This formulation is intended to represent ensemble-style MV approaches, but it is indeed highly inefficient as each sample incurs the full forward-pass cost of the model, making MV more expensive than RT under this implementation.
>
> To avoid this confound, our analysis focuses on more efficient MV variants such as parallel sampling and Tree-of-Thought–style generation. These approaches use a single model invocation (or a shared forward pass) to produce multiple candidate outputs in parallel and then select a final answer. This reflects how MV-style techniques are typically deployed in practice and allows for a fairer comparison with RT in terms of energy consumption.
>
> [1] Self-Consistency Improves Chain of Thought Reasoning in Language Models

---

### Meta-Review · Area_Chair_ZvpJ · 2026-01-06

**Summary:**

The reviewers expressed the following key concerns:

C1. The experimental investigation needs to be considerably more thorough in order to support the general claims that are made. (a) Only one batch size is investigated. (b) Only two types of TTC methods are examined and there is not a clear justification for why they are selected and why they are assumed to fully represent all candidate choices. (c) Only investigates Qwen models with a DeepSeek-R1/RT methodology. (d) Important implementation details are not provided.

C2. The presentation of the results lacks clarity and there is no clear reporting of variability. As an example, Figure 1 shows lines between points that back-track on one another. It’s not clear what the lines are supposed to represent (the connected points are the performances of different models). Each point on this graph represents an average over 4 different tasks, but there is no indication of the variability across task, either in terms of accuracy or energy.

Although one reviewer assigned this paper a score of 10, the review did not provide an in-depth discussion of the results nor clearly explain why the experimental investigation was exemplary with groundbreaking insights. Indeed, the review even suggested that “analyses… could be deepened”, “figures are … not well explained”, and “methodological details…are lacking”. These are exactly the issues that other reviewers raised in assigning scores of 2 and 4. As a result, although I have taken this review and score into account when making a recommendation, the score of 10 does not align with the content of the review.

**Reviewer Concerns:**

Since the response did not provide a revision of the paper, the concerns raised above were not adequately addressed to the extent that the reviewers would have been satisfied by the changes (no changes were made).

The response did not provide substantial, clear details about any additional results that satisfactorily addressed the reservations summarized in C1 and C2.

The presentation of results remained unclear with no depiction of variability.

The paper constitutes an experimental investigation. Without novel methodological or theoretical content, there is an expectation for rigorous, extensive experiment that definitely supports hypotheses and claims. This study contains too many limitations - a single batch size, only two TTC methods, only one LLM family.

**Reviewer Scores:**

The authors did not provide a revised version of the paper. This significantly diminishes the probability of any reviewer changing scores.

Reviewer GeTs. VERY UNLIKELY TO CHANGE. The reviewer raised issues C1 and C2. With no clear additional results, particularly concerning batch size, and no revision to make the presentation of results clear, it is very unlikely that the reviewer would change the score.

Reviewer sXVz. VERY UNLIKELY TO CHANGE. The reviewer raised weaknesses summarized in C1 and C2. Given that the reviewer already assigned a score of 6, and identified several weaknesses that were not clearly addressed in the response, it is very unlikely that the score would change.

Reviewer U7Rn. Already a score of 10; unlikely to lower the score.

Reviewer KEDZ. UNLIKELY TO CHANGE. This review was the most critical and raised multiple points summarized in C1 and C2. Since these were not clearly addressed by any supporting or more extensive results, and there was no paper revision to make the presentation clearer, a change of score is unlikely. Some more minor issues were clarified in the response, so it is possible that the score may have been increased to 4.

---

### Decision · Program_Chairs · 2026-01-26

Reject